# Asparagine Metabolism in Tumors Is Linked to Poor Survival in Females with Colorectal Cancer: A Cohort Study

**DOI:** 10.3390/metabo12020164

**Published:** 2022-02-09

**Authors:** Xinyi Shen, Yuping Cai, Lingeng Lu, Huang Huang, Hong Yan, Philip B. Paty, Engjel Muca, Nita Ahuja, Yawei Zhang, Caroline H. Johnson, Sajid A. Khan

**Affiliations:** 1Department of Chronic Disease Epidemiology, Yale School of Public Health, Yale University, New Haven, CT 06510, USA; xinyi.shen@yale.edu (X.S.); lingeng.lu@yale.edu (L.L.); 2Department of Environmental Health Sciences, Yale School of Public Health, Yale University, New Haven, CT 06510, USA; yupingcai@sioc.ac.cn (Y.C.); h.huang003@gmail.com (H.H.); hong.yan@yale.edu (H.Y.); zhangyw@cicams.ac.cn (Y.Z.); 3Interdisciplinary Research Center on Biology and Chemistry, Shanghai Institute of Organic Chemistry, Chinese Academy of Sciences, Shanghai 200032, China; 4Department of Surgery, Memorial Sloan Kettering Cancer Center, New York, NY 10065, USA; patyp@mskcc.org (P.B.P.); mucae@mskcc.org (E.M.); 5Division of Surgical Oncology, Department of Surgery, Yale University School of Medicine, New Haven, CT 06510, USA; nita.ahuja@yale.edu; 6Department of Surgery, Yale University School of Medicine, New Haven, CT 06510, USA; 7National Cancer Center, National Clinical Research Center for Cancer, Cancer Hospital, Chinese Academy of Medical Sciences and Peking Union Medical College, Beijing 100021, China

**Keywords:** metabolomics, colorectal cancer, prognosis, asparagine metabolism

## Abstract

The interplay between the sex-specific differences in tumor metabolome and colorectal cancer (CRC) prognosis has never been studied and represents an opportunity to improve patient outcomes. This study examines the link between tumor metabolome and prognosis by sex for CRC patients. Using untargeted metabolomics analysis, abundances of 91 metabolites were obtained from primary tumor tissues from 197 patients (N = 95 females, N = 102 males) after surgical colectomy for stage I-III CRC. Cox Proportional hazard (PH) regression models estimated the associations between tumor metabolome and 5-year overall survival (OS) and recurrence-free survival (RFS), and their interactions with sex. Eleven metabolites had significant sex differences in their associations with 5-year OS, and five metabolites for 5-year RFS. The metabolites asparagine and serine had sex interactions for both OS and RFS. Furthermore, in the asparagine synthetase (ASNS)-catalyzed asparagine synthesis pathway, asparagine was associated with substantially poorer OS (HR (95% CI): 6.39 (1.78–22.91)) and RFS (HR (95% CI): 4.36 (1.39–13.68)) for female patients only. Similar prognostic disadvantages in females were seen in lysophospholipid and polyamine synthesis. Unique metabolite profiles indicated that increased asparagine synthesis was associated with poorer prognosis for females only, providing insight into precision medicine for CRC treatment stratified by sex.

## 1. Introduction

According to the U.S. Centers for Disease Control and Detection (CDC), colorectal cancer (CRC) was the third leading cause of cancer-related deaths in the United States in 2021 [1]. Prognosis is partly dependent on stage at presentation according to the TNM classification system based on the American Joint Committee on Cancer (AJCC) [2] and the Union for International Cancer Control (UICC) [3]. Additional clinicopathologic variables can also impact survival, such as primary tumor location [4,5], body mass index (BMI) [6], and diabetes conditions [7]. 

The application of biomarkers in a clinical context has the potential to improve prognosis based on gene mutations [8,9], microRNAs [10], and DNA methylation [11,12]. During the last decade, cancer studies have benefited from the implementation of metabolomics [13]. CRC metabolomic studies have shown the correlation of certain metabolite profiles to cancer survival and recurrence using biological samples of urine and tissue [14]. Some plasma metabolites are associated with recurrence in stage II and III patients [15], showing the potential value of metabolomics in finding prognostic biomarkers of CRC. 

In addition, sex-related differences in CRC prognosis have become increasingly important in cancer research [16,17]. Female patients have a higher prevalence of right-sided colon cancer (RCC), which is associated with poorer overall survival (OS) [18,19]. However, male CRC patients have a survival disadvantage in most subgroups by anatomic location and age [20]. At present, female and male patients do not receive different sex-specific therapies. 

Previously, our untargeted metabolomics analyses in a large series of CRC tumor tissues revealed sex-specific metabolic subphenotypes that could influence tumor progression and response to therapeutics [21]. Important metabolic pathways in these subphenotypes included asparagine synthesis, energy production (glycolysis and pentose phosphate pathway (PPP)), lysophospholipid synthesis, methionine metabolism, and polyamine synthesis [21]. Particularly, female patients with RCC showed increased asparagine levels [21], which can be produced intracellularly by an ATP- and glutamine-dependent reaction catalyzed by asparagine synthetase (ASNS) [22]. By examining The Cancer Genome Atlas (TCGA) data, we discovered that high *ASNS* expression was associated with poorer OS for female CRC patients [21]. Additionally, increased *ASNS* expression has been linked to aggressive tumors and poorer prognosis in other cancer types [23,24]. These findings suggest that sex plays a vital role in CRC prognosis together with the covariates of metabolism and anatomic tumor location. A clear understanding of these sex-specific metabolic differences represents a golden opportunity to prolong the survival of CRC patients.

In this study, we examine whether there is a link between CRC tumor metabolites and prognosis stratified by sex of the patient. This is the first study to reveal that sex-related differences exist in primary tumor metabolite profiles that are associated with OS and recurrence-free survival (RFS) in patients who have undergone curative intent colectomy. We also identify metabolic pathways which are associated with poor survival only in female CRC.

## 2. Results

### 2.1. Population Characteristics

Baseline characteristics of 197 patients (N = 102 males, N = 95 females), including 5-year OS and 5-year RFS, are provided in Table 1. Median follow-up time since date of surgery was 74.8 months (0.1–169.2). Older age, administration of chemotherapy, and advanced clinical stage were inversely related to OS (Table 1). Each subgroup by anatomic tumor location and clinical stage, pathological information (*KRAS* mutation, *BRAF* mutation and microsatellite instability status), demographic characteristics, including race/ethnicity, are displayed in Appendix A. Prognosis among different subgroups is shown in Appendix A.

### 2.2. Sex-Specific Differences in the Associations between Individual Metabolite Abundances and CRC Prognosis

A total of 18 metabolites were significantly associated with OS in males or females or in both sexes (Appendix A), adjusted for anatomic location, clinical stage, chemotherapy history, and age, similarly, 25 metabolites had significant correlations with RFS (Appendix A). Only carnitine and hypoxanthine remained significantly associated with RFS for males after false discovery rate (FDR) adjustment. 

Multivariate Cox PH models on single metabolites and their associations with OS or RFS revealed dichotomous findings by sex. Figure 1 presents the metabolites (adenosine, asparagine, citrulline, glycerol 3-phosphate (Gro3P), lysophosphatidylcholine (LysoPC)(16:0), ornithine, succinate, threonine, UDP-D-glucose, uracil, and xanthosine) that had sex differences in their associations with CRC prognosis (*P*_interaction_ < 0.05). Among these 11 metabolites, succinate was significantly associated with improved OS for female patients (HR_OS_ = 0.35, 95% CI = 0.12–0.99; *p* = 0.047), while it was associated with poorer OS for males (HR_OS_ = 1.91, 95% CI = 1.23–2.96, *p* = 0.004) (Figure 1A). Five metabolites (argininosuccinic acid, asparagine, creatinine, hypoxanthine, and serine) showed significance for RFS and sex-specific differences (Figure 1B). Of these metabolites, asparagine exhibited sex differences for both OS and RFS (*P*_interaction,OS_ = 0.03, *P*_interaction,RFS_ = 0.007), and similarly for serine (*P*_interaction,OS_ = 0.04, *P*_interaction,RFS_ = 0.01) (Appendix A). Asparagine was significantly associated with better CRC prognosis in males (HR_OS_ = 0.72, 95% CI = 0.54–0.96, *p* = 0.03; HR_RFS_ = 0.74, 95% CI = 0.56–0.97, *p* = 0.03), while its associations with worse prognosis in females trended towards significance (*P*_OS_ = 0.13, *P*_RFS_ = 0.09, Appendix A). Similarly, serine was significantly associated with better prognosis in males (HR_OS_ = 0.55, 95% CI = 0.37–0.81, *p* = 0.002. HR_RFS_ = 0.57, 95% CI = 0.40–0.83, *p* = 0.003), while it has no significant associations with prognosis in females (Appendix A). 

Additional analysis using cases with complete information of *KRAS* mutation, *BRAF* mutation and microsatellite instability status showed that 15 metabolites were associated with OS and 3 metabolites were associated with RFS with significant *P*_interaction_ (Appendix C, Table A1 and Table A2). Asparagine was associated with poor RFS in females but not males with a *P*_interaction,RFS_ close to significance (Appendix C, Table A2).

### 2.3. Sex-Specific Differences in CRC Prognosis Associated with Asparagine Synthesis

We examined sex-specific associations between the ASNS-catalyzed asparagine synthesis pathway and CRC prognosis (Appendix A). Significant interactions with sex were detected in associations between asparagine abundance and both OS and RFS (*P*_interaction,OS_ = 0.02, *P*_interaction,RFS_ = 0.003) as summarized as Model 1, where asparagine abundance was treated as a continuous variable (Table 2). In Model 1, asparagine was significantly associated with poor OS (HR_OS_ = 6.39, 95% CI = 1.78–22.91, *p* = 0.004) and poor RFS (HR_RFS_ = 4.36, 95% CI = 1.39–13.68, *p* = 0.01) only in females, while asparagine was only significantly associated with better OS only in males (HR_OS_ = 0.57, 95% CI = 0.36–0.91, *p* = 0.02) after adjustment for clinical stages, chemotherapy history, age, anatomic location, and other crucial metabolites involved in ASNS expression (Table 2). 

An additional multivariate Cox PH regression model was built; Model 2, where asparagine abundance was dichotomized to low levels and high levels. There was a significant interaction between sex and dichotomized asparagine levels for RFS and a similar interaction that was approaching significance for OS (Table 2) (*P*_interaction,OS_ = 0.052, *P*_interaction,RFS_ = 0.03). 

Figure 2 displays the Cox adjusted survival curves for dichotomized asparagine levels based on Model 2, illustrating that female CRC patients with a high level of ASNS-catalyzed asparagine had significantly poorer OS (Figure 2A) and poorer RFS (Figure 2C) compared with female patients with a low level. Strikingly, levels of asparagine were not associated with OS (Figure 2B) or RFS (Figure 2D) for male CRC patients.

ASNS is induced by the PI3K-AKT-mTOR pathway and mutated *KRAS* [21]. Within our study cohort, we had information of *KRAS* status for 161 patients by *KRAS* status. Therefore, we performed a further analysis similar to Model 1 on this subset of patients (Appendix C, Table A3 and Table A4), and asparagine still showed similar sex-specific associations with both OS and RFS.

### 2.4. Sex-Specific Differences in CRC Prognosis Associated with Glycolysis and the Pentose Phosphate Pathway

Considering glycolysis and the PPP metabolism, the association between Gro3P and OS for females trended towards significance (*p* = 0.08), and there was no association in males (*p* = 0.76) (Appendix A). Gro3P was significantly associated with a lower risk of 5-year recurrence only for males (HR_RFS_ = 0.27, 95% CI = 0.08–0.88, *p* = 0.03), though no sex differences were detected (*p*_interaction_ = 0.25) (Appendix A).

### 2.5. Sex-Specific Differences in CRC Prognosis Associated with Lysophospholipid Synthesis

We analyzed the relationships between CRC prognosis and important metabolites involved in lysophospholipid synthesis. The interactions between sex and LysoPC (16:0), LysoPC (18:1), lysophosphatidylethanolamine (LysoPE) (18:0), and LysoPE (22:5) in the association with OS were significant (Appendix A). In the multivariate analysis on OS for lysophospholipid synthesis by sex, LysoPC (16:0) was associated with poor OS only for females (HR_OS_ = 2.07, 95% CI = 1.06–4.07, *p* = 0.03), while it was not significantly associated with OS for males (*p* = 0.32) (Appendix A). In the models of RFS for lysophospholipid synthesis by sex, LysoPE (22:5) presented a robust trend toward a significant sex difference in its association with RFS (*P*_interaction_ = 0.052, Appendix A). LysoPE (22:5) was associated with increased risks of 5-year recurrence of CRC for females (HR_RFS_ = 5.38, 95% CI = 1.02–28.42, *p* = 0.048), while it was associated with reduced risk of recurrence for males (HR_RFS_ = 0.10, 95% CI = 0.02–0.62, *p* = 0.01) (Appendix A).

### 2.6. Sex-Specific Differences in CRC Prognosis Associated with Methionine Metabolism

Analysis of methionine metabolism for all patients indicated that serine favored better OS (HR_OS_ = 0.56, 95% CI = 0.34–0.92, *p* = 0.02) and RFS (HR_RFS_ = 0.49, 95% CI = 0.29–0.83, *p* = 0.008) (Appendix A). After further stratification by sex, serine was associated with better OS in males (HR_RFS_ = 0.37, 95% CI = 0.19–0.72, *p* = 0.004) and improved RFS in males as well (HR_RFS_ = 0.49, 95% CI = 0.24–0.98, *p* = 0.04), while its associations with CRC prognosis in females were not significant (Appendix A). The interaction between serine and sex in terms of OS and RFS were both significant (*P*_interaction,OS_ = 0.04, *P*_interaction,RFS_ = 0.045, Appendix A). Methionine had a significant sex interaction (Appendix A), but it was not significantly associated with OS or RFS in either sex (Appendix A). The additional metabolites in this pathway did not have significant sex interactions (Appendix A).

### 2.7. Sex-Specific Differences in CRC Prognosis Associated with Polyamine Synthesis

Lastly, we examined metabolites involved in polyamine synthesis. Arginine had a marked trend towards a significant sex interaction with OS (*p*_interaction_ = 0.07), and a significant sex interaction for RFS (*p*_interaction_ = 0.04) (Appendix A). Further stratified by sex, arginine was associated with worse OS only for females (HR_OS_ = 5.63, 95% CI = 1.33–23.78, *p* = 0.02), while its association with OS in males was not significant (*p* = 0.87) (Appendix A). For RFS, arginine was associated with higher risks of 5-year CRC recurrence only for females (HR_RFS_ = 3.22, 95% CI = 1.14–9.13, *p* = 0.03) while it had no significant association with RFS for males (*p* = 0.24) (Appendix A). Citrulline suggested significant sex interactions with both OS and RFS (*p*_interaction,OS_ = 0.002, *p*_interaction,RFS_ = 0.02) (Appendix A): it had a significant association with poorer OS for females (HR_OS_ = 2.57, 95% CI = 1.04–6.33, *p* = 0.04) but not for males (*p* = 0.18), while it was not significantly associated with RFS in either sex (Appendix A). Other critical metabolites in polyamine synthesis did not have significant sex interactions (Appendix A).

## 3. Discussion

To our knowledge, this is the first study to show that sex-specific differences in the CRC tumor metabolome are linked to prognosis based on data from one of the largest metabolomics study cohorts. It is also the first application of a novel approach to examine metabolic pathways and their associations to CRC prognosis. By applying state-of-the-art mass spectrometry to fresh frozen specimens acquired operatively in a large colorectal biorepository, we show that females have a distinct metabolite profile, underscored by asparagine metabolism, which correlates with prognosis after surgical colectomy. This suggests that females and males have unique colorectal tumor metabolism, which contributes to sex-specific clinical outcomes. 

Our study revealed that 11 tumor metabolites were associated with CRC 5-year OS and five metabolites for 5-year RFS, and these associations were sex-specific. Asparagine and serine were associated with both 5-year OS and 5-year RFS in a sex-specific manner. Further examination of asparagine within the de novo asparagine synthesis pathway (catalyzed by ASNS), showed consistently poor OS and RFS with high asparagine abundance for females. This supports previous findings revealing poor female survival and high *ASNS* expression from the TCGA [21]. In addition, we previously hypothesized that asparagine is linked to tumor aggressiveness in females with RCCs as it correlates with increased threonine and serine uptake, which indicates aggressive tumor phenotypes [21]. ASNS-catalyzed asparagine production has also been shown to be crucial for *in vitro* cancer cell proliferation by amino acid homeostasis [25], and the availability of asparagine in vitro and in vivo propagates the metastatic progression of breast cancer [26]. 

In CRC, SRY-box 12 (*SOX12*) expression promotes cell proliferation and metastasis, and it facilitates *ASNS* expression [27]. In addition, expression of the mutant *KRAS* gene correlates with a marked decrease in aspartate levels and increased asparagine levels due to upregulated *ASNS* expression, which indicated that ASNS might be a novel therapeutic target for *KRAS*-mutant CRC [28,29]. Moreover, SLC25A22 serves as an essential metabolic regulator for CRC progression [30] by promoting the synthesis of aspartate-derived amino acids (asparagine) in *KRAS*-mutant CRC cells [31]. In our study, *KRAS* information was incomplete but further analysis on patients with complete *KRAS* records can be found in Appendix C (Table A1 and Table A2), where asparagine showed similar sex-specific associations with both OS and RFS. Furthermore, increased dietary asparagine in animals promotes metastatic progression in breast cancer, and L-asparaginase treatment or dietary asparagine restriction inhibits metastasis [32]. Hence, both internally produced asparagine and external exposure to asparagine through diet may influence CRC tumor progression. However, a review study pointed out that some studies are contradictory regarding whether *ASNS* expression favors or inhibits CRC progression [33], while our findings indicate that such inconsistency may attribute to sex differences.

As mentioned, serine was associated with both 5-year OS and RFS in a sex-specific manner. Our data showed that serine favored longer OS and RFS in males but was independent of prognosis in females. Serine metabolism has been reported to contribute to CRC metabolism and growth [34] and enzymes involved in serine synthesis such as phosphoglycerate dehydrogenase (PHGDH) [35] are found in higher levels in colonic tumor tissue than paired normal tissue [36]. These studies did not consider sex differences that might lend to a potential sex-specific phenomenon.

Our results indicated that sex significantly modified the associations between lysophospholipid synthesis and CRC prognosis. LysoPC (16:0) was associated with worse OS for females in both individual metabolite analysis and the model for lysophospholipid synthesis. Our previous study identified that multiple LysoPCs and LysoPEs were upregulated in female patients with RCC (stage I) only, suggesting that the higher level of lysophospholipids in women with RCC would promote fatty acid supply that is essential for cancer cell growth at early stages [21]. It has also been discussed that sex-specific estrogen regulation may inhibit CRC cell survival by suppressing triglyceride biosynthesis, a vital lipid marker related to CRC progression [17]. 

Lastly, sex interacts with the associations between polyamine metabolism and CRC prognosis. Arginine and citrulline were associated with worse RFS and OS, respectively, for females but not for males. A previous study investigated the L-arginine/nitric oxide (NO) pathway in CRC by clinical stage, location of a primary tumor, and sex, and they observed elevated arginine levels in LCCs compared with RCCs, and a drop in arginine levels in the early postoperative period in females exclusively [37]. Arginine and citrulline are important parts of the urea cycle [38], where ornithine is produced and metabolized into polyamines by ornithine decarboxylase (ODC) [39]. ODC is a target for inhibition by difluoromethylornithine (DFMO) (an anti-polyamine therapy for colorectal neoplasia) [40], therefore, some of the polyamine precursors could be important for polyamine treatment responses, where sex can play a role. Moreover, there are NCI-funded cooperative group CRC chemoprevention trials targeting polyamine synthesis (PACES/SWOG) [41], and a review study showed that methionine dietary restriction can suppress tumor formation, potentially through reducing the availability of decarboxylated SAM which is a cofactor for spermidine (a polyamine) production [42]. Therefore, key metabolic pathways may have great applicability in sex-specific treatment or preventative interventions for CRC.

Although all female patients in our analysis were over 55 years and sex hormones might not be able to alter the associations between tumor metabolome and CRC prognosis, the observed sex differences could be attributed to other factors. Sex differences in metabolism are known to exist throughout life-course, as reflected by changes in amino acids level at birth [43], fluctuation of lipid profile of males before and after puberty, with larger sex differences at older ages [44]. Sex chromosomes such as the gene forkhead box P3 (*FOXP3*) located on the X chromosome have been linked to CRC [45]. In addition, there is evidence that sex differences exist before puberty when sex hormone levels rise, and life-course exposure to estradiol can result in elevated DNA and CpG island methylation, as well as demethylation which could result in genetic mutations at later life [46]. Life-course exposure to sex hormones can alter microbiome diversity which would have an impact on microbial metabolism in the colon [47]. Sex differences in the tumor microenvironment such as hypoxia, drug metabolism, and angiogenesis have been shown to impact sex-specific immune responses and oncogenes in CRC [17].

Some of the study limitations include the number of events in the cohorts compared to sample size, and the multiple categories such as clinical stage, sex group, anatomic location, which might weaken statistical power. In addition, prognostic factors such as *KRAS* [48] and *BRAF* mutational status [49], and microsatellite instability status [50] were incomplete in our data; thus, they were not considered in survival analyses for metabolic pathways to avoid less statistical power. Therefore, future studies from an independent cohort with more comprehensive molecular profiling will be valuable. Further, metabolomic analysis of normal colorectal tissues that surround the tumor would also enhance the current findings.

In conclusion, we apply a novel methodology using tumor tissue to identify sex-specific metabolic pathways associated with CRC prognosis in a surgical cohort. We show that tumor metabolite levels have different associations with patient prognosis by sex, in particular ASNS-catalyzed asparagine production that is linked to poor prognosis only in females. Our study shows that sex-specific tumor metabolism is an important factor for understanding differences in patient prognosis and potentially differences in treatment responses. Accordingly, we recommend that the design of precision medicine approaches consider both tumor metabolism and sex to obtain more robust findings in clinical trials and improve patient outcomes.

## 4. Materials and Methods

### 4.1. Sample Collection and Metabolite Measurements

Fresh frozen tumor tissue samples were acquired after curative intent colectomy for patients (N = 197) with RCCs (cecum, ascending colon, and hepatic flexure) or LCCs (splenic flexure, descending colon, sigmoid, rectosigmoid) with stage I-III CRC (1991–2000). No patients received neoadjuvant chemotherapy. Rectal cases had different biology mechanisms to the colon, and transverse tissue could not be accurately assigned as right or left in original, and thus they were not included. Stage IV tissues would have been treated with chemotherapy before surgery, which were therefore excluded. Finally, abundances of 91 metabolites were obtained as published previously [21]. Metabolites were extracted and analyzed by hydrophilic interaction liquid chromatography mass spectrometry (HILIC-MS) and reverse phase liquid chromatography mass spectrometry (RPLC-MS)-based metabolomics using a UPLC system (H-Class ACQUITY, Waters Corporation, Massachusetts, United States) with a quadrupole time-of flight (QTOF) mass spectrometer (Xevo G2-XS QTOF, Waters Corporation, Milford, Massachusetts, United States) as previously described [21]. Other pathological clinical records such as *KRAS* mutation, *BRAF* mutation and microsatellite instability status were available but incomplete. More information about and explanations the timeline of sample collection, and metabolite measurements are presented in Appendix B. Yale University IRB determined that this study was not considered Human Subjects Research and did not require IRB review (IRB/HSC no. 1612018746).

### 4.2. Statistical Analysis

We included age, sex, anatomic primary tumor location, chemotherapy history, and clinical stage as covariates. Missing values of metabolites abundances were replaced using median imputation. Multivariable Cox proportional hazard (Cox PH) regression models were constructed to evaluate the associations between prognosis with individual metabolite abundance per standard deviation (SD) increase on a log_2_-scale, adjusted for covariates in either all patients or stratified by sex. Assessment of associations between sex-specific metabolic pathways and prognosis was conducted by multivariate Cox PH models that included important metabolites involved in these pathways. Considering statistical power, cases with complete pathological information (*KRAS* mutation, *BRAF* mutation and microsatellite instability status) were only sufficient to analyze associations between prognosis and individual metabolites but were not to analyze metabolic pathways. Sex-specific difference was examined in multivariate Cox PH models by including an interaction between sex and the metabolite. The 5-year OS and 5-year RFS were calculated. Due to the absence of death events in female patients at clinical stage I, we recoded the clinical stages I and II as “early stage”, and III as “late stage”. Patients with adjuvant postoperative chemotherapy were coded as having chemotherapy history. Survival analyses were conducted using package “survival” in R (version 4.0.4). A two-sided *p* value less than .05 was considered statistically significant.

## Figures and Tables

**Figure 1 metabolites-12-00164-f001:**
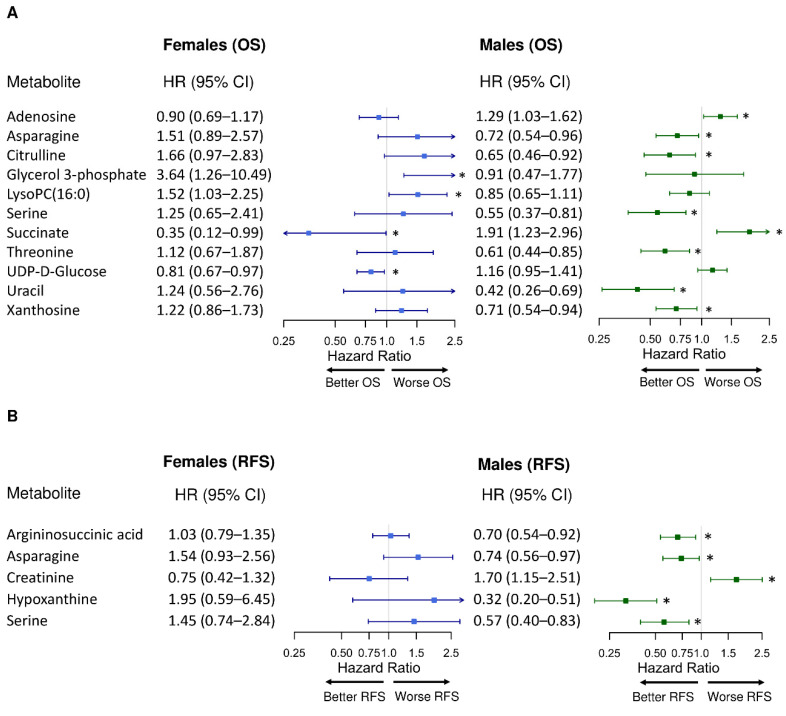
Sex-specific differences in the associations between individual metabolite and CRC prognosis. (**A**) 5-year overall survival (OS) and (**B**) 5-year recurrence-free survival (RFS). LysoPC: lysophosphatidylcholine. All metabolites were observed with significant sex interactions (*P*_interaction_ < 0.05). Hazard ratio (HR) for each metabolite (log_2_-transformed abundance) was calculated by multivariate Cox PH regression adjusted for anatomic location, chemotherapy history, clinical stage, and age (continuous). Error bars represent 95% confidence intervals (CIs). The 95% CIs marked with asterisks indicate significant associations between the metabolite and the corresponding prognosis (*p* values < 0.05): A metabolite with HR < 1 was associated with better prognosis; a metabolite with HR > 1 was associated with worse prognosis. The x-axes are log-scaled.

**Figure 2 metabolites-12-00164-f002:**
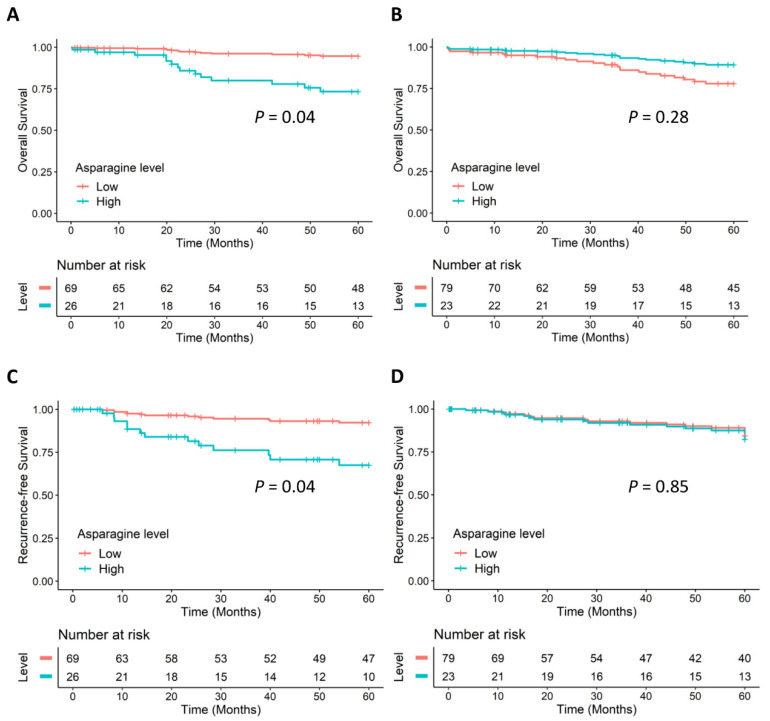
Cox adjusted survival curves of ASNS-catalyzed asparagine. (**A**) OS of females, (**B**) OS of males, (**C**) RFS of females, and (**D**) RFS of males. Results were based on Model 2 (Table 2) using multivariate Cox PH regression, adjusted for anatomic location, chemotherapy history, clinical stages, age (continuous), and other crucial metabolites involved in ASNS-catalyzed asparagine synthesis (aspartate, glutamate, glutamine, and AMP were treated as continuous variables and were log_2_ transformed). *p* values for the associations between dichotomized asparagine levels and the prognosis in Model 2.

**Table 1 metabolites-12-00164-t001:** Demographic Characteristics and Clinical Factors.

Characteristics		No. of Patients	5-Year OS			5-Year RFS		
			Deaths, No.	Rate, % ^a^	*p* ^b^	Cases, No.	Rate, % ^a^	*p* ^b^
Age at diagnosis, y	≤60	19	2	88.9	0.048	3	84.2	0.21
	61–69	64	9	83.4		14	72.4	
	70–79	81	15	78.7		12	81.4	
	≥80	33	11	61.5		1	96.3	
Sex, n	Male	102	23	74.3	0.18	17	77.5	0.48
	Female	95	14	83.2		13	84.0	
Clinical stage, n	I	47	3	92.5	0.001	5	88.4	0.09
	II	86	13	82.4		11	82.0	
	III	64	21	63.5		14	73.8	
Chemotherapy, n	Yes	66	18	68.5	0.03	15	73.1	0.03
	No	131	19	83.7		15	85.0	
Anatomic tumor location, n	Left	99	17	81.2	0.42	19	77.2	0.23
	Right	98	20	75.6		11	85.3	

^a^ Survival rates were calculated by the Kaplan–Meier estimation method. ^b^
*p* value of Log–rank test.

**Table 2 metabolites-12-00164-t002:** Multivariate analysis of asparagine synthesis catalyzed by asparagine synthetase (ASNS).

Prognosis	Sex	Model 1 ^a^			Model 2 ^b^		
		HR (95% CI) ^c^	*p*	*p* _interaction_ ^d^	HR (95% CI) ^c^	*p*	*p* _interaction_ ^d^
OS	Females	6.39 (1.78–22.91)	0.004	0.02	5.68 (1.06–30.61)	0.04	0.052
	Males	0.57 (0.36–0.91)	0.02		0.46 (0.11–1.84)	0.27	
RFS	Females	4.36 (1.39–3.68)	0.01	0.003	4.89 (1.07–22.39)	0.04	0.03
	Males	0.96 (0.61–1.50)	0.86		1.15 (0.27–4.80)	0.85	

^a^ Asparagine abundance was considered as a continuous variable (log_2_ transformed). ^b^ Asparagine abundance was dichotomized to low levels (≤75% percentile of asparagine abundance among all patients) (reference) and high levels (>75% percentile). ^c^ Hazard ratio for asparagine abundance in the corresponding multivariate Cox PH model, adjusted for anatomic location, clinical stages, chemotherapy history, age (continuous), and other crucial metabolites involved in ASNS-catalyzed asparagine synthesis (aspartate, glutamate, glutamine, and AMP were treated as continuous variables and were log_2_ transformed). ^d^ *p* values for the interaction between sex and asparagine.

## Data Availability

The untargeted metabolomics data underlying this article are available at www.ebi.ac.uk/metabolights (accessed on 23 January 2022) MTBLS1122, MTBLS1124, MTBLS1129, MTBLS1130.

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
