# Peer review of "Asparagine Metabolism in Tumors Is Linked to Poor Survival in Females with Colorectal Cancer: A Cohort Study"

_metabolites, 2022, doi:10.3390/metabo12020164_

Round 1

Reviewer 1 Report

This is a very interesting, even inspiring research study. Nothing to complain but one fact - which is in my opinion very important and thus a major point of criticism, which should be addressed in order to improve the manuscript even further:

  • please add the data on BRAF-mutational status and MSI. According to the description in materials and methods it seems possible that frozen tissue might be left to allow for these analyses. If it is impossible to get for all or further cases, give this information for those cases available AND discuss the missing data as a clear limitation of the study for known reasons of prognostic impact of BRAFmut as well as MSI.

Author Response

Dear reviewer,

We appreciate the positive comments from the reviewer. Admittedly, BRAF mutation status and MSI status are important for CRC prognosis, however, we only have information on BRAS and MSI for a subset of patients from the clinical records. We did carry out an analysis using the subset of samples for which we had the data, however, this sample size is smaller than cases with complete KRAS mutation information, which were used for analyzing sex-specific associations between ASNS-catalyzed asparagine synthesis. In the new version of the manuscript, we add an explanation of this limitation in the discussion. We also include a revised Table S1 with more information on these variables and an updated Appendix B that includes additional analysis using cases with complete information of KRAS mutation, BRAF mutation and MSI status.

Reviewer 2 Report

Manuscript ID: metabolites-1566669

Title: Asparagine metabolism in tumors is linked to poor survival in females with colorectal cancer: A cohort study

 Xinyi Shen et al.                                    

 The authors show sex-specific differences in the metabolome colorectal cancer-related and they discuss that these differences are linked to prognosis.

The study is well conducted, however, I have some comments:

Comments:

  1. Different epidemiological studies report gender differences in the incidence and site distribution of colorectal cancer. Here, the authors should better discuss why female sex affects prognosis. Most of the female patients included in the study are elderly patients, in whom the effects of sex hormones are negligible;
  2. The study of the metabolome in the "normal" colorectal tissue surrounding the tumor could provide further evidence to confirm the current results, given also the small sample size of this study;
  3. The Table 1 needs to be improved in the graphic form

Reviewer 3 Report

The interplay between the sex-specific differences in tumor metabolome and colorectal cancer (CRC) prognosis represents an opportunity to improve patient outcomes. In the manuscript the authors based on samples from 197 patients (95 females and 102 males) showed that unique metabolite profiles indicated increased asparagine synthesis was associated with poorer prognosis for females. The conclusions are interesting. Although some limitations of research exist, they were discussed in the text.

According to minor issues:

  1. In Table 2 I do not see letter "d" described in the Table legend, please correct.
  2. In the text you should use full name Figure, not "Fig"

Author Response

We thank the reviewer for the positive feedback on our study and great suggestions on the formatting. Please see the updated version of our manuscript where Table 2 is corrected and the wording for figures is modified.

Round 2

Reviewer 1 Report

My concern has been adequately considered.

One minor mistake has been introduced in the modified text:

Sex chromosomes such as the gene forkhead box
protein P3 (FOXP3) located on the X chromosome have been linked to CRC [45].